# Evolutionary Adaptation by Repetitive Long-Term Cultivation with Gradual Increase in Temperature for Acquiring Multi-Stress Tolerance and High Ethanol Productivity in *Kluyveromyces marxianus* DMKU 3-1042

**DOI:** 10.3390/microorganisms10040798

**Published:** 2022-04-09

**Authors:** Sornsiri Pattanakittivorakul, Tatsuya Tsuzuno, Tomoyuki Kosaka, Masayuki Murata, Yu Kanesaki, Hirofumi Yoshikawa, Savitree Limtong, Mamoru Yamada

**Affiliations:** 1Life Science, Graduate School of Science and Technology for Innovation, Yamaguchi University, Yamaguchi 753-8515, Japan; sornkiri@gmail.com (S.P.); tkosaka@yamaguchi-u.ac.jp (T.K.); 2Department of Biological Chemistry, Faculty of Agriculture, Yamaguchi University, Yamaguchi 753-8515, Japan; tg019vh@yahoo.co.jp; 3Research Center for Thermotolerant Microbial Resources, Yamaguchi University, Yamaguchi 753-8315, Japan; muratam@yamaguchi-u.ac.jp; 4Research Institute of Green Science and Technology, Shizuoka University, Suruga-ku, Shizuoka 422-8529, Japan; kanesaki.yuh@shizuoka.jp; 5Department of Bioscience, Tokyo University of Agriculture, Tokyo 156-8502, Japan; hiyoshik@nodai.ac.jp; 6Department of Microbiology, Faculty of Science, Kasetsart University, Chatuchak, Bangkok 10900, Thailand; fscistl@ku.ac.th

**Keywords:** repetitive long-term cultivation with gradual increases in temperature (RLCGT), *K. marxianus* DMKU 3-1042, evolutionarily adapted mutants, thermotolerant ethanol-fermenting yeast, multi-stress tolerance, genomic and transcriptomic analyses

## Abstract

During ethanol fermentation, yeast cells are exposed to various stresses that have negative effects on cell growth, cell survival, and fermentation ability. This study, therefore, aims to develop *Kluyveromyces marxianus*-adapted strains that are multi-stress tolerant and to increase ethanol production at high temperatures through a novel evolutionary adaptation procedure. *K. marxianus* DMKU 3-1042 was subjected to repetitive long-term cultivation with gradual increases in temperature (RLCGT), which exposed cells to various stresses, including high temperatures. In each cultivation step, 1% of the previous culture was inoculated into a medium containing 1% yeast extract, 2% peptone, and 2% glucose, and cultivation was performed under a shaking condition. Four adapted strains showed increased tolerance to ethanol, furfural, hydroxymethylfurfural, and vanillin, and they also showed higher production of ethanol in a medium containing 16% glucose at high temperatures. One showed stronger ethanol tolerance. Others had similar phenotypes, including acetic acid tolerance, though genome analysis revealed that they had different mutations. Based on genome and transcriptome analyses, we discuss possible mechanisms of stress tolerance in adapted strains. All adapted strains gained a useful capacity for ethanol fermentation at high temperatures and improved tolerance to multi-stress. This suggests that RLCGT is a simple and efficient procedure for the development of robust strains.

## 1. Introduction

*Kluyveromyces marxianus* is a thermotolerant yeast that has beneficial properties for industrial ethanol fermentation, including efficient production of ethanol at high temperatures, high growth rate, short doubling times, weak glucose repression, and the ability to assimilate various sugars, such as glucose, xylose, and sucrose, that are present in various raw materials [1,2,3,4,5]. Studies on *K. marxianus* have been increasing rapidly, and the high potential of *K. marxianus* for industrial applications has been reviewed [6,7,8,9,10]. Among the *K. marxianus* strains studied, DMKU 3-1042, one of the most thermotolerant strains that was isolated in Thailand, has been extensively analyzed [1,7,11], and its complete genome sequence has been determined by transcriptomic analysis under four different growth conditions [12].

Attention has been given to bioethanol worldwide as an alternative to fossil fuels, as it is one of the cleanest and most renewable energy sources [13,14]. In the ethanol industry, the temperature inside fermenters rises during the fermentation process, and the increase in temperature suppresses cell growth, cell viability and ethanol production, thus increasing production costs [13,15]. Accordingly, fermenting microbes that can withstand high temperatures, such as *K. marxianus*, are beneficial and also allow high-temperature fermentation (HTF). HTF has several advantages, including reduced cooling costs, reduced risk of contamination, and reduced amounts of hydrolysis enzymes for simultaneous saccharification and fermentation [5,16,17,18]. It is also beneficial for fermentation in tropical countries and under global warming conditions. In addition to temperature tolerance, fermenting microbes are required to be resistant to various stressors present during fermentation, such as ethanol, acetic acid, osmotic stress, and oxidative stress [13,19,20]. Therefore, HTF that uses stress-tolerant microbes can reduce the total cost of ethanol production, bring cheaper ethanol compared to commercial materials, and provide benefits for industrial application [10,21].

Various strategies such as adaptation, mutagenesis, sexual breeding, and genetic engineering have been reported for the amelioration of yeast properties, such as enhanced stress tolerance and ethanol production [22,23,24,25]. Of these, the use of various stress adaptation techniques has resulted in improvement of the properties of several yeast strains [4,14,24,26,27,28,29]. For example, Saini et al. [4] adapted a *K. marxianus* strain to a whey permeate medium containing a high concentration of lactose for 65 days and acquired one adapted strain with an ethanol yield of 17.5%. Zhang et al. [28] obtained an *S. cerevisiae* strain via a 266 nm laser radiation treatment followed by repeated cultures in the presence of ethanol, and the strain produced a larger amount of ethanol than that produced by the parental strain. In most cases, yeast cells were repeatedly subjected to short-term cultures at fixed temperatures under certain stress conditions; therefore, most adapted strains acquired predominantly a single enhanced property (see Appendix A). However, since cells are exposed to various types of stress during fermentation, improvements of multiple properties are desired. Repetitive long-term cultivation with gradual increases in temperature (RLCGT) was performed by repeatedly cultivating yeast at a low temperature and then by transferring it to a higher temperature and cultivating repeatedly in the same temperature. Yeast cells can adapt to high temperatures during exposure with stepwise increased temperatures.

In this study, in order to develop a simple and effective adaptation procedure for cells to be exposed to various stressors, we examined RLCGT close to the critical high temperature for growth, which is expected to provide various stresses such as those from metabolites including ethanol or organic acids, by-products formed by chemical reactions, nutrient starvation, high temperatures, and oxidative stress in addition to large changes in substrate sugar concentration. Therefore, each cultivation step may optimize resistance to not only temperature but also to other various stresses. The adapted *K. marxianus* strains obtained by RLCGT exhibited increased tolerance to several stresses and enhanced ethanol production compared to the parental strain. Genomic and transcriptomic analyses of the strains were performed.

## 2. Materials and Methods

### 2.1. Yeast Strains

*K. marxianus* DMKU 3-1042, which is a thermotolerant ethanol-fermenting yeast that was isolated in Thailand [1], and its derivatives were used in this study. The yeast strains were preserved in YPD medium (10 g L^−1^ yeast extract, 20 g L^−1^ peptone and 20 g L^−1^ glucose) supplemented with 100 g L^−1^ glycerol at −80 °C.

### 2.2. Determination of Cell Growth and Viability

The pre-culture was prepared by transferring a single colony of 18 h culture on a YPD agar plate into 30 mL of YPD medium in a 100 mL Erlenmeyer flask, followed by incubation in a rotary shaker at 160 rpm for 18 h at 30 °C. The pre-culture was inoculated into YPD medium or YP medium containing 160 g L^−1^ glucose at the initial optical density at 660 nm (OD_660_) of 0.1, and the culture was carried out at 30–47 °C under a shaking condition at 100 or 160 rpm. Cell growth was determined by measuring the OD_660_ on a UV-VIS spectrophotometer (Shimadzu, Japan). Cell viability was determined as colony-forming units (CFU) by counting the number of colonies on YPD agar plates. Samples were diluted with sterile distilled water and were spread on plates, and colonies were counted after 48-h of incubation at 30 °C. Experiments were performed in triplicate.

### 2.3. Determination of Fermentation Parameters

During cultivation, samples were taken and subjected to an analysis of fermentation parameters. Glucose, acetate, and ethanol were analyzed by using a high-performance liquid chromatography (HPLC) system (Hitachi, Japan) consisting of a Hitachi Model D-2000 Elite HPLC system Manager, column oven L-2130, pump L-2130, auto-sampler L-2200, and RI detector L-2490, equipped with a GL-C610H-S gel pack column at 60 °C with 0.5 mL min^−1^ eluent of 0.1% phosphoric acid. pH was measured by using a twin pH meter (Horiba, Japan).

### 2.4. Evolutionary Adaptation by RLCGT

For improvement of the stress tolerance of *K. marxianus* DMKU 3-1042, RLCGT was carried out. The pre-culture was inoculated into ten test tubes including 3 mL of YPD medium and was incubated at 40 °C and 160 rpm for 7 days. The adaption started at 40 °C because *K. marxianus* DMKU 3-1042 was found to be unable to survive under long-term cultivation at 41 °C. After that, 1% of the culture was transferred into a fresh medium (other inoculation sizes were not tested), and cultivation was repeated under the same conditions. After two cultivations at 40 °C, 1% of the culture at the second time was transferred into a fresh medium and cultivated at 41 °C and 160 rpm for 7 days. Cultivation was repeated with a gradual increase in temperature from 40 °C to 45 °C (Appendix A). Cultivation was performed twice at each temperature. Finally, the RLCGT cultures that survived at 45 °C were streaked on YPD plates, and a single colony from each tube was isolated as an adapted strain. This method introduces random mutations and cannot reproduce the same characteristics as the improved strains.

### 2.5. Characterization of Adapted Strains

The tolerance of the adapted strains to various stresses found during the fermentation process was investigated. To prepare an inoculum, cells were grown in YPD medium at 30 °C for 18 h and recovered by low-speed centrifugation. The cells were suspended with sterile distilled water, adjusted to OD_660_ of 1.0, and ten-fold sequentially diluted. Five microliters of cell suspension was spotted onto the surface of YPD agar plates supplemented with various stress materials; the agar plates were adjusted to pH 3 or supplemented with 0.3% (*v*/*v*) (52.45 mM) acetic acid, 0.1% (*v*/*v*) (26.50 mM) formic acid, 8% (1.37 M) (*v*/*v*) ethanol, 15 mM furfural, 15 mM hydroxymethylfurfural (HMF), 0.1% (*w*/*v*) (0.70 mM) vanillin, multiple inhibitors (0.3% (*v*/*v*) acetic acid, 15 mM furfural and 0.15% (*w*/*v*) (1.05 mM) vanillin or 0.15% (*v*/*v*) (26.23 mM) acetic acid, 7.5 mM furfural and 0.075% (*w*/*v*) (0.52 mM) vanillin), 35% (*w*/*v*) (0.30 M) glucose, or 5 mM hydrogen peroxide (H_2_O_2_). YP agar plates supplemented with 10% (*w*/*v*) (0.10 M) xylose were also used. The plates after spotting were incubated at 30 °C, 37 °C, 40 °C, and 45 °C for 48 h. Moreover, growth and fermentation ability of the adapted strains were examined in YPD medium supplemented with 0.2% or 0.3% acetic acid, 10 mM or 15 mM furfural, or multiple inhibitors (0.15% acetic acid, 7.5 mM furfural and 0.075% vanillin) and/or a high concentration of glucose (8% (0.07 M), 12% (0.10 M) or 16% (0.14 M)) at high temperatures for 6 h to 48 h. All experiments were performed in triplicate.

### 2.6. Preparation of Genomic DNA, Genomic Sequencing, and Genome Mapping Analysis

The genomic DNA of adapted strains was extracted as described previously [30] from cells grown in YPD medium for 18 h under a shaking condition at 30 °C and was further purified using a Genomic-tip 20 kit (Qiagen, Hilden, Germany) according to the manufacturer’s instructions. Genome sequencing was carried out by a massively parallel sequencer (MiSeq; Illumina KK, Tokyo, Japan) as reported previously [31]. The sequenced reads were screened by a quality score higher than the Phred score of 30 and were trimmed 12 bases from the 5′ end and 20 bases from the 3′ end. Truncated reads less than 150 bases or with ambiguous nucleotides were excluded from further analysis. Accession numbers of all sequence data are DRR305124-DRR305128.

For genome mapping analysis, the reference genome sequence of *K. marxianus* DMKU 3-1042 (GenBank acc. No: AP012213-AP012221) was downloaded from NCBI ftp site, https://ftp.ncbi.nlm.nih.gov/ (accessed on 5 March 2022). Mutation sites were searched for by read mapping using CLC Genomics Workbench version 7.5 (Qiagen, Venlo, Netherland) with the following parameters: match score: 1, mismatch cost: 2, insertion/deletion cost: 3, length fraction: 0.7, and similarity fraction: 0.9. The filter settings for SNP and Indel calling were the same as those use in a previous study [32]. All mutations in coding and non-coding regions in all adapted mutants were confirmed by the Sanger method [33] after amplification of in-dividual regions by PCR using primers listed in Appendix A. Physiological functions of mutated genes were analyzed by a BLAST search at NCBI (https://www.ncbi.nlm.nih.gov (accessed on 5 March 2022)) or with the STRING database (https://string-db.org (accessed on 5 March 2022)).

### 2.7. RNA-Seq Analysis

RNA for RNA-Seq analysis was prepared as described previously [7]. The parental and ACT001 strains were inoculated into YPD medium, and pre-culture was carried out with a rotary shaker at 30 °C and 160 rpm for 18 h. The pre-culture was inoculated into 30 mL of YPD medium in a 100 mL Erlenmeyer flask at OD_660_ of 0.1, and culture was carried out at 45 °C for 8 h under a shaking condition at 160 rpm. The cells were harvested by centrifugation at 5000 rpm for 5 min at 4 °C and subjected to an RNA preparation process. RNA was prepared by a modified procedure on the basis of the procedure reported previously [12]. The RNA samples then were subjected to RNase-free DNase treatment. All RNA samples were purified by using an RNeasy plus mini kit (QIAGEN, Hilden, Germany) according to the protocol provided by supplier.

The purified RNA samples were analyzed on an Illumina MiniSeq at the Research Center of Yamaguchi University. The detailed procedure for RNA-Seq has been de-scribed previously [34]. All of the data were deposited under accession numbers DRR305122 and DRR305124. The sequencing results were analyzed using CLC genomic workbench version 10.1.1. All mapped reads at exons were counted, and the numbers were converted to unique exon reads. The unique exon reads from two biological rep-licates of ACT001 were compared to those of the parental strain.

Gene expression profiles of ACT001 and the parental strain were compared to find differentially expressed genes (DEGs) based on unique exon read values from CLC genomic workbench outputs using DESeq2 R package [35]. The resulting *p*-values were adjusted using Benjamin–Hochberg’s method for controlling the false discovery rate. Genes with adjusted *p* values less than 0.05 and log_2_ (fold change) values greater than 1 or lower than −1 were assigned as significant DEGs.

## 3. Results

### 3.1. Effects of Long-Term Cultivation on K. marxianus DMKU 3-1042

In general, adaptive laboratory evolution depends on selection pressure for accumulating adapted strains in the designed cultivation. In this study, as selection pressure, we used long-term cultivation at high temperatures, which provides various stressors that prevent cell growth or cause cell damage, such as organic acids excreted into the culture medium, limitation of nutrients, and high temperatures. Before the adaptation experiments, several parameters of *K. marxianus* DMKU 3-1042 during long-term cultivation in YPD medium including 2% glucose were compared between two different temperatures for 1 week (Figure 1). The results showed that cell growth and cell viability at 45 °C were lower than those at 30 °C. Glucose was almost completely utilized within 12 h at both temperatures (Figure 1d). The maximum ethanol concentrations were 7.53 g L^−1^ and 6.46 g L^−1^ at 30 °C and 45 °C, respectively (Figure 1e). During fermentation, ethanol was converted to acetic acid, and its accumulation was significantly high at 45 °C compared to that at 30 °C (Figure 1f). The concentration of acetate continued to rise, reaching 5.44 g L^−1^ at 168 h, consistently lowering the pH of the medium (Figure 1c). The viability at 45 °C approached zero at 120 h, whereas that at 30 °C was maintained, though the level decreased after 120 h (Figure 1b). The decline in viability at 45 °C began when the pH fell below the pKa of acetic acid, a condition under which acetic acid enters cells in a non-dissociated form by facilitated diffusion, causing cytotoxicity. The production of acetic acid that causes low pH may be a major cause of reduced viability, but other factors such as high temperature, limitation of nutrients, and/or other by-product accumulation could be directly or indirectly associated with the earlier decline in cell viability at 45 °C. These findings suggest that various stressors that cause a decline in viability by long-term cultivation at high temperatures are useful selection pressures for this yeast.

### 3.2. Adaptive Laboratory Evolution of K. marxianus DMKU 3-1042 by RLCGT

In order to obtain strains with multi-stress tolerance, *K. marxianus* DMKU 3-1042 was subjected to RLCGT while gradually increasing the temperature from 40 °C to 45 °C. Cells were inoculated into 10 tubes containing 3 mL YPD and cultivated at 40 °C for 7 days, and a portion of the culture in each tube was inoculated into a tube containing fresh YPD medium, cultivated at 40 °C for 7 days, and so on (Appendix A). For 7 days, cells were exposed to several stresses such as ethanol, acetate, other by-products, and nutrient starvation in addition to high temperature. These combinations may significantly reduce survival. The temperature was increased by 1 °C every 2 weeks. No growth was observed in 6 tubes at 44 °C, and no growth was observed in the remaining 4 tubes above 45 °C. After repetitive cultivation for 12 weeks, a portion of each culture in the remaining 4 tubes was spread on YPD agar plates to obtain ACT001, ACT002, ACT003, and TML001, which were used in the following experiments.

### 3.3. Fermentation Ability of Adapted Strains at High Temperatures

In industrial ethanol production, heat generated by fermentation is one of the most serious problems for achieving stable and efficient fermentation. The effects of temperature on growth and fermentation parameters of the four adapted strains were thus examined in YP medium containing 160 g L^−1^ glucose at 40 °C, 42 °C, 45 °C, and 47 °C (Figure 2). At 40 °C and 42 °C, three adapted strains, ACT001, ACT002, and ACT003, utilized glucose almost completely within 24 h, whereas glucose remained in the culture of TML001 and the parental strain. Maximal ethanol concentrations of ACT001, ACT002, and ACT003 at 40 °C were 69.7 g L^−1^, 68.5 g L^−1^, and 70.3 g L^−1^, respectively, whereas maximal ethanol concentrations of TML001 and the parental strain were 46.9 g L^−1^ and 49.6 g L^−1^, respectively. As a result of the evaporation of ethanol increasing with increasing temperature [16], the amount of ethanol produced at high temperatures may be underestimated. At higher temperatures of 45 °C and 47 °C, glucose in the medium remained even in ACT strains after 48 h of fermentation, and the four adapted strains produced higher ethanol concentrations than that produced by the parental strain. Maximal ethanol concentrations produced by these strains were in the range of 41.5–56.2 g L^−1^ (Table 1). At 42 °C and 45 °C, higher ethanol concentrations and lower acetate accumulation were achieved by the four adapted strains in comparison to the parental strain. The ethanol productivities and ethanol yields of ACT001, ACT002, and ACT003 were almost the same at each temperature and were higher than or equal to those of TML001 (Table 1).

### 3.4. Characterization of the Adapted Strains

During the fermentation process, yeast cells are exposed to various stresses that have negative effects on cell viability and ethanol production [13,19,25], and it is desirable that yeast cells are resistant to these stresses. We thus compared the stress tolerance of the four adapted strains that had experienced stressful conditions during the adaptation process with that of the parental strain (Figure 3). Their growth was examined on YPD agar plates supplemented with 0.3% (*v*/*v*) acetic acid, 0.1% (*v*/*v*) formic acid, or an adjusted pH of 3 at different temperatures up to 45 °C. ACT001, ACT002, and ACT003 grew well except for in 0.3% (*v*/*v*) acetic acid at 45 °C, but TML001 and the parental strain did not grow (Figure 3a). When examined on YPD agar plates supplemented with 8% (*v*/*v*) ethanol, which also inhibits cell growth [24], all of the adapted strains grew better than did the parental strain at 40 °C, and TML001 showed much stronger growth (Figure 3b). When examined on YPD agar plates supplemented with 15 mM furfural or 15 mM HMF, which are inhibitors found in lignocellulosic hydrolysate and browning reaction products [29,36], all of the adapted strains grew better than did the parental strain at 40 °C and 45 °C (Figure 3b). The tolerance to vanillin as a phenolic compound [37] was also examined, showing that all adapted strains were more resistant to 0.1% (*w*/*v*) vanillin than the parental strain (Figure 3c).

Further experiments were carried out with a YPD liquid medium supplemented with 0.2% (*v*/*v*) and 0.3% (*v*/*v*) acetic acid under a shaking condition at 40 °C. In the case of 0.2% (*v*/*v*) acetic acid, ACT001, ACT002, and ACT003 showed much better growth than that of TML001 and the parental strain, and they completely utilized glucose and produced the highest concentration of ethanol within 12 h, whereas TML001 and the parental strain completely utilized glucose and produced the highest concentration of ethanol at 48 h. In the case of 0.3% (*v*/*v*) acetic acid, only ACT strains grew and utilized glucose and produced ethanol (Figure 4). These results are consistent with the results of the above-described experiments using agar plates, indicating that ACT strains are resistant to acetic acid. Moreover, all adapted strains except ACT002 showed better growth and ethanol production than the parental strain in a YPD medium supplemented with 10 mM furfural at 45 °C (Appendix A). These results combined with the results obtained from experiments with formic acid, furfural, and HMF suggest that the adapted strains have acquired abilities to withstand multiple stressors.

Considering that the stresses in biofuel production from lignocellulosic biomass are synergistic, the effects of multiple inhibitors including acetate, furfural, and vanillin were examined (Figure 3c). The synergistic effects of inhibitors were seen when each inhibitor in the experiments with multiple inhibitors was added at the same concentration as used in the experiments with a single inhibitor (Figure 3a,b). Under these conditions, all the adapted strains did not grow. However, when the concentrations of inhibitors were reduced, all adapted strains except for TML001 grew, but the parental strain did not grow at 30 °C and 45 °C. These findings suggest that ACTs have acquired resistance to multiple stresses. Furthermore, ACT003, which was most tolerant to acetic acid (Figure 4) and furfural (Appendix A), was subjected to fermentation experiments in a YPD medium containing 0.15% acetate, 7.5 mM furfural, and 0.075% vanillin in the presence of 8–16% glucose at 40 °C (Appendix A). ACT003 showed much better growth and ethanol production than the parental strain.

To examine the effects of osmotic stress and oxidative stress, which are known to accumulate in cells at high temperatures [20], growth experiments were carried out on YP agar plates supplemented with 10% (*w*/*v*) xylose or 35% (*w*/*v*) glucose and supplemented with 5 mM H_2_O_2_. In the former experiments, the growth of all the adapted strains was almost the same as that of the parental strain, except that the growth of TML001 was weaker in the presence of 35% glucose at 45 °C (Appendix A). Additionally, in the latter experiments, ACT strains exhibited a level of growth similar to that of the parental strain, but the growth of TML001 was slightly weaker at 45 °C (Appendix A).

### 3.5. Mutation Points of Adapted Strains

Next-generation sequencing was performed to find mutations in the four adapted strains, and candidate mutations were further confirmed by direct sequencing. The results showed that ACT001, ACT002, ACT003, and TML001 strains have two, four, one, and seven mutation points, respectively (Table 2). There were no shared mutations in ACT strains, though they exhibited a similar tolerance phenotype against several stresses tested. The two mutations in ACT001 are the following: the insertion of one nucleotide in the coding region of *KLMA_10738* for a PH domain-containing protein as an orthologue of YHR131C in *S. cerevisiae*, causing a frame shift mutation, and the deletion of one nucleotide in the non-coding region. All mutations in ACT002 and ACT003 are single nucleotide polymorphisms (SNPs) either in the coding or non-coding regions, causing one missense and two synonymous mutations in the case of ACT002. The missense mutation occurs in *GAL1* for galactokinase, and the synonymous mutations occur in *ALY2* for UPF0675 protein as an orthologue of YJL084C in *S. cerevisiae* and in *KLMA_40563* for ATP-dependent protease La. YHR131C contains a PH domain that occurs in a wide range of proteins involved in intracellular signaling or as constituents of the cytoskeleton [38,39,40,41]. There are six SNPs and one triple-nucleotide insertion in TML001. Of these, four are missense or nonsense mutations in coding regions, and one causes a single amino acid insertion. The missense mutations occur in *FSH3* for the family of serine hydrolase 3 and in *TNA1* for a high-affinity nicotinic acid transporter; the nonsense mutations occur in *KLMA_40326* for an orthologue of YPL014W in *S. cerevisiae* and in *SVL3* for styryl dye vacuolar localization protein 3; and the amino acid insertion mutation occurs in *PMA1* for plasma membrane ATPase. YPL014W (Cip1) is a novel negative regulator of cyclin-dependent kinase [41].

### 3.6. Transcriptome Analysis

The three strains ACT001, ACT002, and ACT003 had similar phenotypes of multi-stress tolerance and higher ethanol production but had obtained distinct mutations, indicating that it is difficult to understand the mechanism of expression of similar phenotypes only by genome mutation analysis. We thus carried out transcriptome analysis of ACT001 as a representative strain and of the parental strain as a control by RNA-Seq using total RNAs prepared from cells grown in a YPD medium at 45 °C for 8 h. Reads per kilobase of exon per million (RPKM) of each gene were estimated as a transcript abundance. The difference of each gene in ACT001 strain from that in the parental strain was reflected as the ratio of the RPKM value in ACT001 strain to that in the parental strain. To further explore the transcriptional changes, analysis of differentially expressed genes (DEGs) based on the RNA-Seq data was conducted. DEGs showed significant changes at the transcription level with log_2_ (fold change) > 1 and log_2_ (fold change) < −1. A total of 7 genes were significantly up-regulated in ACT001 strain (Table 3), and 10 genes were significantly down-regulated (Table 3).

Of the significantly up-regulated genes (Table 3), *ICL1*, *CIT3*, and *ADY2* may be related to acetic acid tolerance by maintaining a low level of intracellular acetate. The former two genes enhance the TCA cycle or glyoxylate cycle to assimilate acetic acid, and the latter exports acetate. Consistent with the assimilation enhancement, the acetate accumulation level in ACT001 was lower than that in the parental strain at high temperatures (Figure 2). *ICL1* encodes isocitrate lyase, which catalyzes the cleavage of isocitrate to succinate and glyoxylate and which is a key enzyme for the glyoxylate cycle to utilize two carbon compounds, such as acetate [42]. *CIT3* encodes the dual specificity mitochondrial citrate and methylcitrate synthase, which catalyzes the condensation of acetyl-CoA and oxaloacetate to form citrate, and which catalyzes that of propionyl-CoA and oxaloacetate to form 2-methylcitrate, respectively, in *S. cerevisiae* [43]. *ADY2* is an acetate transporter that has a key role in acetic acid sensitivity, and disruption of *ADY2* abolishes the active transport of acetate in *S. cerevisiae* [44]. The up-regulation of *ADY2* may reduce the intracellular acetate level. Therefore, it is possible that the enhanced expression of these three genes is responsible for acid tolerance in ACT001. Given that three ACT mutants exhibited similar levels of acetic acid tolerance and similar glucose utilization patterns at high temperatures, ACT002 and ACT003 can enhance the TCA cycle or glyoxylate cycle as well as export the activity of acetate.

Four genes, *HAK1*, *ZRT2*, *CDR4*, and, for transporters including a high-affinity potassium transporter, a low-affinity zinc transporter, an ATPase-coupled transporter, and a putative allantoate transporter were down-regulated. Two genes, *SNZ3* and *SNO3*, for probable pyridoxal 5′-phosphatae biosynthesis were also down-regulated. The relationship between these negative regulations and stress resistance is not clear.

## 4. Discussion

There are many reports showing improvement of yeasts via evolutionary adaptation [4,14,24,26,28,29,45,46]. In most of them, cells were exposed to a single stress, for example, a high concentration of ethanol, by repetitive cultivation in a short cycle, such as 1 to 2 days at a fixed temperature (Appendix A). The procedure applied in this study is relatively simple and not laborious because of the transfer to a fresh medium once a week, which gives a chance to enrich multi-stress-resistant mutants and to obtain characteristically different mutants if cultivated in multiple test tubes. We successfully acquired 4 different mutants from cultivation with 10 test tubes in this study.

Long-term cultivation causes nutrient starvation and toxic compounds to accumulate in the medium. *K. marxianus* DMKU 3-1042 was found to accumulate acetate at high temperatures (Figure 1), which may be responsible for a marked decrease in cell viability after 36 h. It suggests that this yeast tends to convert ethanol to acetic acid and/or has a weak metabolism with acetic acid at high temperatures. High temperatures may be not a primary factor affecting cell viability because it can grow even at 48 °C [16], but high temperatures may lead to the accumulation of acetate. On the other hand, the accumulation of acetic acid is relatively low at low temperatures, and the decline of cell viability after 120 h could be due to limitation of nutrients. Therefore, the acquisition of aceticacidresistant mutants of ACTs at high temperatures is reasonable. Notably, the accumulation levels of acetate were lower than that of the parental strain in the medium containing 160 g L^−1^ of glucose at 40 °C to 45 °C. Given the high ethanol production in ACT mutants, their ethanol-to-acetic acid conversion activity may be relatively low, or acetic acid metabolic activity may be high as expected from the results of the transcriptome analysis. TML001, which is sensitive to acetic acid like the parental strain, may have a distinct strategy for survival in the presence of acetic acid. The mutant somehow kept a lower level of acetate compared to the other adapted mutants as seen in the medium containing 160 g L^−1^ of glucose at 40 °C to 45 °C. Notably, *K. marxianus* DMKU 3-1042 as well as adapted strains exhibited a stronger resistance to high temperatures (up to 48 °C) on agar plates [16]. It is assumed that only the surface cells of colonies on agar plates are exposed to oxidative stress, which increases with rising temperature [47], and individual cells in liquid media are directly exposed to the stress. The thermal stability of *K. marxianus* DMKU 3-1042 is one of the most important application criteria for different applications, and this property is crucial for the preparation strategy of the strain or its stabilization. The stability of *K. marxianus* DMKU 3-1042 is beneficial for a wide range of applications in synthesis processes, which currently are in great demand from the industrial point of view [10,21].

By RLCGT, adapted mutants were acquired at a relatively high frequency (4 mutants/10 tubes). One of the reasons may be a significant reduction in viability at each 7-day cultivation step. Gradual rising temperatures may reduce survival because, in addition to temperature stress, higher temperatures tend to accumulate more reactive oxygen species (ROS) that damage cells [5,21]. The combinations of various stresses including acetic acid, high temperatures, ROS, ethanol, and other by-products are thought to influence viability, resulting in enrichment of adapted mutants. The other may be due to the gradual accumulation of ethanol, acetic acid, and other by-products at each long-term cultivation step. The high acquisition frequency of adapted mutants and their enhanced capacity of ethanol fermentation at high temperatures demonstrate the usefulness of RLCGT for evolutionary adaptation. This procedure can be applied to improve other microbes once the initial temperature for RLCGT is properly determined. Recently, *Candida tropicalis* X-17 was subjected to RLCGT in the presence of a high concentration of glucose, and the resultant adapted strain increased ethanol production and improved multi-stress tolerance [48].

Spontaneous mutations occur during cultivation, and some of the mutations contribute to adaptation to environmental conditions. Such adaptations are also accompanied by phenotypic alterations [4,49]. In this study, all of the adapted strains exhibited phenotypes of resistance to furfural, HMF, and vanillin in addition to ethanol or acetic acid. The phenotypes were stable rather than transient, which may be conferred by underlying mutations, unlike an adaptive response and tolerance to stress encountered during ethanol fermentation [13,34]. ACT strains were clearly tolerant to acetic acid, formic acid, and low pH compared to TML001 and the parental strain (Figure 3). Acetic acid resistance may have promoted glucose assimilation and ethanol production more than the parental strain (Figure 2). One to seven mutations in coding or non-coding regions were found in the adapted strains, some of which may be responsible for altered phenotypes. Unfortunately, the mutations in the adapted strains were different, and mutations responsible for the common phenotypes were therefore not identified. Mutations causing drastic alterations in protein structure could be related to adaptations under stress conditions, including the frame-shift mutation of *KLMA_10738* in ACT001 and the nonsense mutations of *KLMA_40326* and *SVL3* in TML001. Null mutants of their orthologue genes, *YHR131C*, *YPL014W*, and *SVL3*, in *S. cerevisiae* exhibited an increase in competitive fitness in minimum medium [40], and a resistance to oxidative stress [50], respectively. Therefore, it is assumed that the accumulation of some mutations except for ACT003, not only in coding regions but also non-coding regions, is responsible for a phenotype such as acetic acid resistance. In the future, it is necessary to confirm stress tolerance by constructing individual mutations in the wild-type background.

Nonetheless, transcriptome analysis suggests possible mechanisms of acetic acid tolerance in ACT001 as mentioned above, though the relationship between its genomic mutations and acetic acid tolerance is unclear. Like these adapted strains, adapted strains with multi-stress tolerance were previously isolated in *S. cerevisiae* [14,24]. The mechanism of multi-stress tolerance may be complicated, given previous evidence which suggests that many genes are involved in certain stress resistances [51], that genes required for specific stress tolerance contribute to other stress resistances [52], and that the integration of selected genes can improve tolerance to multiple inhibitors in lignocellulose fermentations [53].

No direct association between up-regulation of those three genes and mutations in ACT001 can be found in the literature or databases. ACT001 was suggested to have two mutations: a frame-shift mutation in *KLMA_10738* for the Pleckstrin homology (PH) domain-containing protein and an SNP in the non-coding region (Table 2). PH domains are small protein modules known for their ability to bind to phosphoinositides, and they are involved in intracellular signaling or in being constituents of the cytoskeleton [38,39,40,54]. The STRING database indicates that *YHR131C* in *S. cerevisiae*, which is the orthologue of *KIMA_10738* in *K. marxianus*, interacts with 10 proteins, including a putative transcription factor required for growth of superficial pseudohyphae. Therefore, it is speculated that the defective mutation of *KLMA_10738* is directly or indirectly related to stress resistance and/or the up-regulation of *ICL1*, *CIT3*, and *ADY2* in ACT001. Further works are needed in the future, including the disruption of *KIMA_10738* and its effects on the expression of these genes, and including the overexpression of these genes and their effects on acetic acid and other stress tolerances.

The four mutants obtained in this study produced higher ethanol concentration than the parental strain in a YP medium containing 160 g L^−1^ at high temperatures under a shaking condition, which provided oxygen and avoided aggregations of cells. Ethanol concentrations in ACT mutants increased by about 20–40% of that of the parental strain (Table 1). The increment was consistent with greater consumption of glucose compared to that of the parental strain. The ratios of increase in ethanol production are higher than or equivalent to those of adapted mutants in several reports. The ratios of increase in ethanol concentration in *S. cerevisiae* SM4 adapted in a YP medium containing 300 g L^−1^ glucose and 50 g L^−1^ ethanol at 34 °C, an adapted strain from *K. marxianus* MTCC1389 in whey permeate containing 200 g L^−1^ lactose at 37 °C, and *S. cerevisiae* YF10-5 adapted in a fermentation medium containing 350 g L^−1^ glucose at 30 °C, were in the range of 16.6% to 29.3% [4,14,28]. Therefore, the adaptation procedure used in this study may allow efficient improvement of ethanol productivity at high temperatures.

## 5. Conclusions

This study showed that (1) RLCGT is an effective evolutionary adaptation procedure and provided evidence that (2) adapted strains have the ability to achieve high ethanol concentrations at high temperatures and are (3) capable of tolerating multiple stresses, especially ACT strains, which are strongly resistant to acetic acid and formic acid. These beneficial properties are useful for industrial ethanol fermentation using lignocellulosic biomass as a substrate.

## Figures and Tables

**Figure 1 microorganisms-10-00798-f001:**
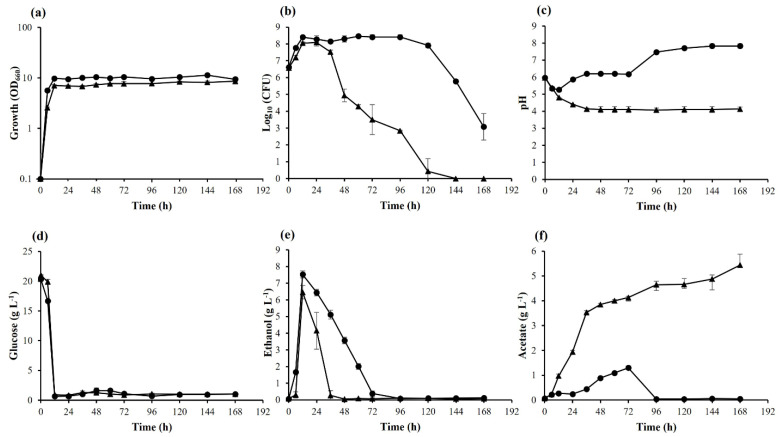
Growth, CFU, and fermentation parameters of *K. marxianus* DMKU 3-1042 in YPD medium at 30 °C and 45 °C. Cells were cultivated in YPD medium at 30 °C (●) and 45 °C (▲) under a shaking condition at 160 rpm for 7 days. Growth (**a**) was determined by measuring OD_660_. CFUs (**b**) were determined by counting colonies on YPD plates incubated at 30 °C. pH (**c**) was measured by using a twin pH meter. Glucose (**d**), ethanol (**e**), and acetate (**f**) were determined by HPLC. Error bars represent ±SD of values from experiments performed in triplicate.

**Figure 2 microorganisms-10-00798-f002:**
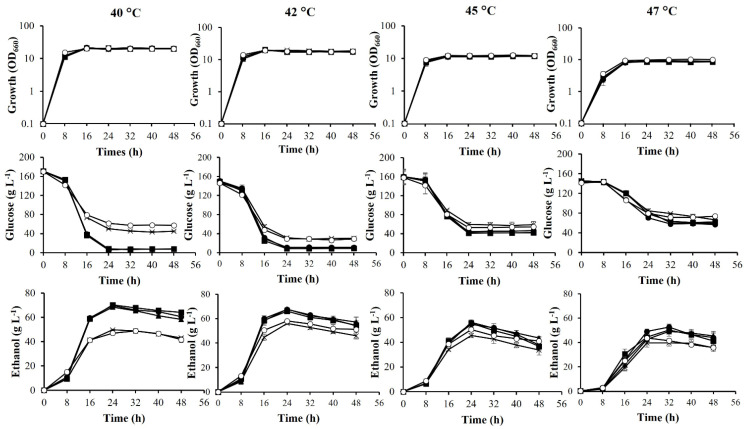
Fermentation of adapted strains in YP medium containing 16% glucose at high temperatures. Wild-type (✕), ACT001 (●), ACT002 (▲), ACT003 (■), and TML001 (◯) strains were cultivated in YP medium containing 16% glucose at 40–47 °C under a shaking condition at 100 rpm, and samples were taken every 8 h until 48 h. Growth was determined by measuring OD_660_. Glucose, ethanol, and acetate were determined by HPLC. Error bars represent ±SD of values from experiments performed in triplicate.

**Figure 3 microorganisms-10-00798-f003:**
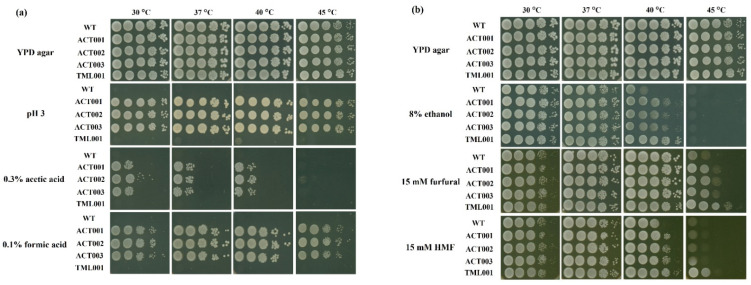
Characterization of adapted strains. Growth of adapted strains on YPD agar plates with pH adjusted to 3 or supplemented with 0.3% acetic acid, 0.1% formic acid (**a**), 8% ethanol, 15 mM furfural, 15 mM HMF (**b**), 0.1% vanillin or multiple inhibitors (0.15% acetate, 7.5 mM furfural and 0.075% vanillin or 0.3% acetate, 15 mM furfural and 0.15% vanillin) (**c**) were compared. The plates were incubated at 30–45 °C for 48 h.

**Figure 4 microorganisms-10-00798-f004:**
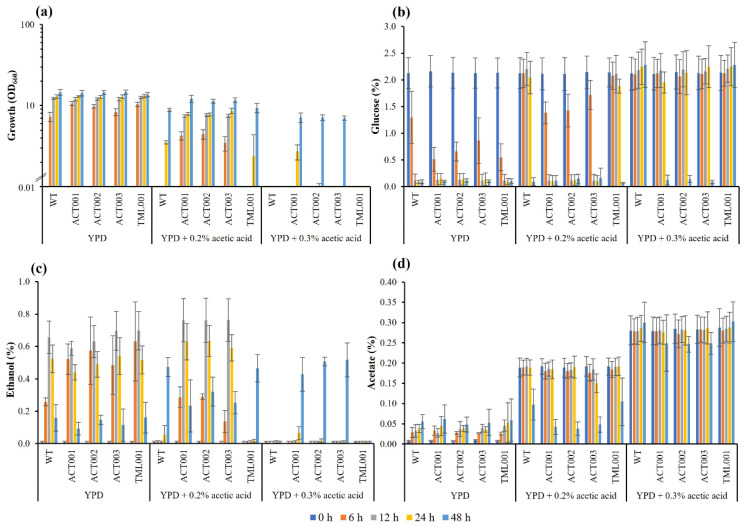
Effects of acetic acid on growth and fermentation parameters of adapted strains at 40 °C. Adapted strains were cultivated in YPD medium supplemented with 0.2% or 0.3% acetic acid at 40 °C under a shaking condition at 160 rpm for 48 h. Growth (**a**) was determined by measuring OD_660_. Glucose (**b**), ethanol (**c**), and acetate (**d**) were determined by HPLC. Error bars represent ±SD of values from experiments performed in triplicate.

**Table 1 microorganisms-10-00798-t001:** Comparison of growth and fermentation parameters among various adapted strains and previously reported strains.

Strains	Temp. (°C)	Medium	Cultivation Time (h)	Growth (OD_660_)	Remaining Sugar (g L^−1^)	Acetic Acid Accumulation (g L^−1^)	Ethanol Production (g L^−1^)	Increased Ethanol (%)	Ethanol Productivity (g L^−1^ h^−1^)	Ethanol Yield (g g^−1^)	Ref
*K. marxianus* DMKU 3-1042											
Wild-type	40	YP + 160 g L^−1^ glucose	24	19.0 ± 0.8	50.1 ± 1.2	1.7 ± 0.1	49.6 ± 0.7	-	2.1 ± 0.0	0.3 ± 0.0	This study
ACT001	40	YP + 160 g L^−1^ glucose	24	19.4 ± 0.7	7.3 ± 2.4	1.5 ± 0.1	69.7 ± 0.7	40	2.9 ± 0.0	0.4 ± 0.0	This study
ACT002	40	YP + 160 g L^−1^ glucose	24	20.3 ± 1.2	5.8 ± 2.0	1.5 ± 0.0	68.5 ± 0.7	38	2.9 ± 0.0	0.4 ± 0.0	This study
ACT003	40	YP + 160 g L^−1^ glucose	24	20.7 ± 1.0	8.1 ± 0.4	1.5 ± 0.0	70.3 ± 0.7	42	2.9 ± 0.0	0.4 ± 0.0	This study
TML001	40	YP + 160 g L^−1^ glucose	24	20.7 ± 0.9	61.5 ± 0.4	1.3 ± 0.1	46.9 ± 0.7	0	2.0 ± 0.1	0.3 ± 0.0	This study
Wild-type	42	YP + 160 g L^−1^ glucose	24	19.6 ± 1.1	31.2 ± 2.2	1.7 ± 0.1	56.1 ± 1.1	-	2.3 ± 0.1	0.4 ± 0.0	This study
ACT001	42	YP + 160 g L^−1^ glucose	24	17.5 ± 1.3	11.3 ± 1.1	1.4 ± 0.0	67.6 ± 0.3	20	2.8 ± 0.0	0.4 ± 0.0	This study
ACT002	42	YP + 160 g L^−1^ glucose	24	17.5 ± 0.9	8.7 ± 1.4	1.5 ± 0.0	67.9 ± 1.2	21	2.8 ± 0.0	0.4 ± 0.0	This study
ACT003	42	YP + 160 g L^−1^ glucose	24	17.0 ± 0.8	8.8 ± 1.4	1.5 ± 0.0	66.2 ± 1.0	18	2.8 ± 0.0	0.4 ± 0.0	This study
TML001	42	YP + 160 g L^−1^ glucose	24	17.9 ± 1.2	29.0 ± 0.8	1.3 ± 0.0	58.1 ± 0.4	3	2.4 ± 0.0	0.4 ± 0.0	This study
Wild-type	45	YP + 160 g L^−1^ glucose	24	12.1 ± 0.6	59.5 ± 4.0	1.6 ± 0.0	45.5 ± 1.8	-	1.9 ± 0.1	0.3 ± 0.0	This study
ACT001	45	YP + 160 g L^−1^ glucose	24	11.3 ± 0.5	44.5 ± 2.0	1.4 ± 0.0	54.5 ± 3.0	20	2.3 ± 0.1	0.3 ± 0.0	This study
ACT002	45	YP + 160 g L^−1^ glucose	24	11.9 ± 1.0	42.8 ± 1.6	1.4 ± 0.0	56.2 ± 1.8	23	2.3 ± 0.1	0.4 ± 0.0	This study
ACT003	45	YP + 160 g L^−1^ glucose	24	11.6 ± 0.6	41.4 ± 1.6	1.4 ± 0.1	55.4 ± 2.0	22	2.3 ± 0.1	0.4 ± 0.0	This study
TML001	45	YP + 160 g L^−1^ glucose	24	12.2 ± 0.7	52.9 ± 2.3	1.4 ± 0.0	50.3 ± 1.7	11	2.1 ± 0.1	0.3 ± 0.0	This study
Wild-type	47	YP + 160 g L^−1^ glucose	32	9.3 ± 1.1	78.9 ± 1.6	1.9 ± 0.1	39.5 ± 2.8	-	1.2 ± 0.1	0.3 ± 0.0	This study
ACT001	47	YP + 160 g L^−1^ glucose	32	8.9 ± 0.9	57.8 ± 1.4	1.8 ± 0.2	52.6 ± 2.5	33	1.7 ± 0.1	0.3 ± 0.0	This study
ACT002	47	YP + 160 g L^−1^ glucose	32	9.0 ± 0.7	60.2 ± 3.2	1.8 ± 0.1	49.0 ± 3.8	24	1.5 ± 0.1	0.3 ± 0.0	This study
ACT003	47	YP + 160 g L^−1^ glucose	32	8.5 ± 0.5	63.5 ± 1.3	1.7 ± 0.1	50.2 ± 1.8	27	1.6 ± 0.1	0.3 ± 0.0	This study
TML001	47	YP + 160 g L^−1^ glucose	32	10.0 ± 0.8	71.4 ± 2.0	1.8 ± 0.1	41.5 ± 1.7	5	1.3 ± 0.1	0.3 ± 0.0	This study
*S. cerevisiae* G85	28	Sugar juice (207.25 g L^−1^ sugar)	480	NR	10.4 ± 0.8	NR	126.6 ± 5.6	-	0.3	0.6	[24]
*S. cerevisiae* G85X-8	28	Sugar juice (207.25 g L^−1^ sugar)	480	NR	7.0 ± 1.0	NR	130.0 ± 4.5	3	0.3	0.6	[24]
*S. cerevisiae* YE0	34	YP + 300 g L^−1^ glucose + 50 g L^−1^ ethanol	72	NR	31.5	NR	106.8	-	1.5	0.4	[28]
*S. cerevisiae* SM4	34	YP + 300 g L^−1^ glucose + 50 g L^−1^ ethanol	72	NR	2.1	NR	138.1	29	1.9	0.5	[28]
*K. marxianus* MTCC1389 (Wild-type)	37	Whey permeate (200 g L^−1^ lactose)	50	NR	0.7 ± 0.0	NR	66.8 ± 0.9	-	1.3 ± 0.0	0.3 ± 0.0	[4]
*K. marxianus* MTCC1389 (Adapted strain)	37	Whey permeate (200 g L^−1^ lactose)	42	NR	0.8 ± 0.0	NR	79.3 ± 0.8	19	1.7 ± 0.1	0.4 ± 0.1	[4]
*S. cerevisiae* Y-1	30	Fermentation medium (350 g L^−1^ glucose)	60	NR	75.7	NR	125.0	-	2.1 ± 0.0	0.3	[14]
*S. cerevisiae* YF10-5	30	Fermentation medium (350 g L^−1^ glucose)	60	NR	5.5	NR	145.8	16.6	2.43 ± 0.1	0.4	[14]

NR: not reported.

**Table 2 microorganisms-10-00798-t002:** Summary of mutations of adapted strains.

Adapted Strain	Gene/Locus_Tag	Product	Region	Ref	Allele	Type	Amino Acid Change
ACT001	KLMA_10738	PH domain-containing protein Orthologue of YHR131C	1552906^1552907	-	G	Insertion	Lle812fs
			47658	T	-	Deletion	Non-coding region
ACT002			1161817	G	A	SNP	Non-coding region
	GAL1	Galactokinase	712698	G	T	SNP	Lue391Phe
	ALY2	UPF0675 protein Orthologue of YJL084C	385614	G	A	SNP	synonymous
	KLMA_40563	ATP-dependent protease La	1269738	T	C	SNP	synonymous
ACT003			1107715	T	C	SNP	Non-coding region
TML001	FSH3	Family of serine hydrolase 3	613476	C	G	SNP	Lle253Met
	PMA1	Plasma membrane ATPase	968552^968553	-	GCT	Insertion	Ala255_Leu256insAla
	TNA1	High-affinity nicotinic acid transporter	1010140	C	G	SNP	Phe467Leu
			1714543	G	A	SNP	Non-coding region
	KLMA_40326	Hypothetical protein Orthologue of YPL014W	767036	G	T	SNP	Glu106 *
			724738	T	G	SNP	Non-coding region
	SVL3	Styryl dye vacuolar localization protein 3	815396	C	A	SNP	Ser629 *

SNPs: Single nucleotide polymorphisms; fs: frame-shift mutation; * stop codon.

**Table 3 microorganisms-10-00798-t003:** Up-regulated (log_2_ > 1) and down-regulated (log_2_ < −1) genes in ACT001 strain.

Locus_Tag	Gene	Log_2_ Fold Change	Product
Up-regulated			
KLMA_70179	*ICL1*	1.90	Isocitrate lyase
KLMA_70444	*CIT3*	1.43	Citrate synthase 3
KLMA_30101	*SPG4*	1.24	stationary phase protein 4
KLMA_20819	*KLMA_20819*	1.20	hypothetical protein
KLMA_60471	*RRT12*	1.12	putative subtilase-type proteinase YCR045C
KLMA_60452	*FBP1*	1.08	fructose-1,6-bisphosphatase
KLMA_20009	*ADY2_1*	1.04	acetate transporter
Down-regulated			
KLMA_50379	*HAK1*	−1.99	high-affinity potassium transporter
KLMA_50489	*ZRT2*	−1.39	zinc-regulated transporter 2
KLMA_10655	*CDR4*	−1.37	ATPase-coupled transporter
KLMA_50332	*SEO1*	−1.33	probable transporter SEO1
KLMA_10677	*MET17*	−1.18	protein MET17
KLMA_30339	*KLMA_30339*	−1.17	ATP synthase subunit b
KLMA_30724	*SNZ3*	−1.17	pyridoxal-5′-phosphate synthase
KLMA_30338	*KLMA_30338*	−1.08	protein ICY2
KLMA_60029	*ACAD11*	−1.01	acyl-CoA dehydrogenase family member 11
KLMA_30726	*SNO3*	−1.00	probable pyridoxal-5′-phosphate synthase subunit

Each strain was performed in duplicate.

## Data Availability

All data are reported in this manuscript.

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
