# Peer review of "Evolutionary Adaptation by Repetitive Long-Term Cultivation with Gradual Increase in Temperature for Acquiring Multi-Stress Tolerance and High Ethanol Productivity in Kluyveromyces marxianus DMKU 3-1042"

_microorganisms, 2022, doi:10.3390/microorganisms10040798_

Round 1
Reviewer 1 Report
This study introduces a novel evolutionary adaptation procedure called “Repetitive Long-term Cultivation with Gradual Increase of Temperature (RLCGT)” to develop adapted strains of yeast (Kluyveromyces marxianus, a thermotolerant ethanol-fermenting yeast)) that are tolerant to multi-stressors with concomitant increase in the production yields of ethanol at fermentation temperatures close to the critical temperature of 48 degree C. The approach makes good sense as it allows the organism to adapt gradually to various stresses including ethanol, acetic acid, other by-products and nutrient starvation, and large changes in substrate sugar concentration, in addition to the higher temperatures. Four adapted strains were obtained and characterized by genome and transcriptome analyses. The approach is simple and a significant departure from other strategies (so the work is original), but most importantly, it works. The study has been meticulously performed and the experiments are well described. I recommend publication of the manuscript as is.

Author Response
Reviewer 1
Comments and Suggestions for Authors
This study introduces a novel evolutionary adaptation procedure called “Repetitive Long-term Cultivation with Gradual Increase of Temperature (RLCGT)” to develop adapted strains of yeast (Kluyveromyces marxianus, a thermotolerant ethanol-fermenting yeast)) that are tolerant to multi-stressors with concomitant increase in the production yields of ethanol at fermentation temperatures close to the critical temperature of 48 degree °C. The approach makes good sense as it allows the organism to adapt gradually to various stresses including ethanol, acetic acid, other by-products and nutrient starvation, and large changes in substrate sugar concentration, in addition to the higher temperatures. Four adapted strains were obtained and characterized by genome and transcriptome analyses. The approach is simple and a significant departure from other strategies (so the work is original), but most importantly, it works. The study has been meticulously performed and the experiments are well described. I recommend publication of the manuscript as is.
Reviewer did not have any comment, so we have not response.

Reviewer 2 Report
The current work focuses on the Evolutionary Adaptation by Repetitive Long-Term Cultivation with Gradual Increase of Temperature for Acquiring Multi-Stress Tolerance and High Ethanol Productivity in Kluyveromyces marxianus DMKU 3-1042. The experimental work appears to have been carried out well. However, a few points deserve attention for further publication. I suggest that it is accepted for publication after the following revisions:
- ABSTRACT: In this study, Evolutionary Adaptation by Repetitive Long-Term Cultivation with Gradual Increase of Temperature for Acquiring Multi-Stress Tolerance and High Ethanol Productivity in Kluyveromyces marxianus DMKU 3-1042. What parameters were optimized on the Evolutionary Adaptation by Repetitive Long-Term Cultivation with Gradual Increase of Temperature for Acquiring Multi-Stress Tolerance and High Ethanol Productivity in Kluyveromyces marxianus DMKU 3-1042? Authors must include numbers with the results found. Stability of the Kluyveromyces marxianus DMKU 3-1042? How much Kluyveromyces marxianus DMKU 3-1042 were utilized to process? Furthermore, what are the conditions of reactions? Temperature, pH, ionic strength, for example. This information should be included in the abstract.
- INTRODUCTION:
- Evolutionary Adaptation by Repetitive Long-Term Cultivation with Gradual Increase of Temperature for Acquiring Multi-Stress Tolerance and High Ethanol Productivity in Kluyveromyces marxianus DMKU 3-1042: This process needs to be explained in the introduction of the manuscript.
- The Evolutionary Adaptation by Repetitive Long-Term Cultivation with Gradual Increase of Temperature for Acquiring Multi-Stress Tolerance and High Ethanol Productivity in Kluyveromyces marxianus DMKU 3-1042: What optimization strategy was used? Why was it used? This information needs to be explained in the introduction of the manuscript.
- The new process presented were compared with a commercial material?? This information must be clear in the introduction.
- What the advantages in Kluyveromyces marxianus DMKU 3-1042? These strategies used should be better explained in the manuscript.
MATERIALS:
- Include the molar concentration of solutions.
METHODS:
- Include the molar concentration of all the chemicals used, the way the methods are presented, not possible reproducibility.
- Preparation procedure: Please include more details, temperature, pH, molar ratio, ionic strength.
- RESULTS AND DISCUSSION:
- The thermal stability to Kluyveromyces marxianus DMKU 3-1042 prepared is one of the most important application criteria for diferent applications. This stability depends on the preparation strategy. It also depends on the stabilization of the Kluyveromyces marxianus DMKU 3-1042. This discussion could be improved. Please include in the manuscript.
- The stability of Kluyveromyces marxianus DMKU 3-1042 its wide application in synthesis processes which nowadays are in a great demand from the point of view of industrial. Please include in the manuscript.
- Did the authors conduct studies with non Kluyveromyces marxianus DMKU 3-1042? How can the authors state the effects of Kluyveromyces marxianus DMKU 3-1042 and non- Kluyveromyces marxianus DMKU 3-1042on the characteristics of the produced biocatalysts? The authors need to explain the results in the manuscript.
- Was determined the full loading of Kluyveromyces marxianus DMKU 3-1042under the optimized conditions? This information must be clear in the manuscript.
- The Kluyveromyces marxianus DMKU 3-1042may experience aggregation (mainly near to the isoelectric point). This may be caused by undesired - interactions where inactivation that can stabilize incorrect in structures. This results must be cleared in the manuscript.
- The optimization of Kluyveromyces marxianus DMKU 3-1042preparation process, the preparations shown having diffusion limitations? Considering the strategy presented in this manuscript. Please, this should be explained in the manuscript. What were the optimum conditions?
- Others factors that cause the loss of durability and stability of the Kluyveromyces marxianus DMKU 3-1042should be explained in the manuscript.
- CONCLUSIONS: The main contributions to the accomplishment of this work must be included in the conclusion. Please, authors must use numbers.
- Please, check all references according to the author's instructions.
- Include more details in the figures (error bars) and tables captions.
- The manuscript must be formatted according to the journal's standards.
Author Response
Reviewer 2
Comments and Suggestions for Authors
The current work focuses on the Evolutionary Adaptation by Repetitive Long-Term Cultivation with Gradual Increase of Temperature for Acquiring Multi-Stress Tolerance and High Ethanol Productivity in Kluyveromyces marxianus DMKU 3-1042. The experimental work appears to have been carried out well. However, a few points deserve attention for further publication. I suggest that it is accepted for publication after the following revisions:
- ABSTRACT:
In this study, Evolutionary Adaptation by Repetitive Long-Term Cultivation with Gradual Increase of Temperature for Acquiring Multi-Stress Tolerance and High Ethanol Productivity in Kluyveromyces marxianus DMKU 3-1042. What parameters were optimized on the Evolutionary Adaptation by Repetitive Long-Term Cultivation with Gradual Increase of Temperature for Acquiring Multi-Stress Tolerance and High Ethanol Productivity in Kluyveromyces marxianus DMKU 3-1042?
According to this comment, we have added the sentence of “Therefore, each cultivation step may optimize resistance to not only temperature but also other various stresses.” in the last paragraph of the introduction section.
Authors must include numbers with the results found.
According to this comment, we have included numbers with the results found as follows. This study showed that (1) RLCGT is an effective evolutionary adaptation procedure and provided evidence that (2) adapted strains acquired have the ability to achieve high ethanol concentrations at high temperatures and (3) are capable of tolerating multiple stresses, especially ACT strains, which are strongly resistant to acetic acid and formic acid. These beneficial properties are useful for industrial ethanol fermentation using lignocellulosic biomass as a substrate. (in the conclusion section)
Stability of the Kluyveromyces marxianus DMKU 3-1042?
According to this comment, we have added the sentences of “Notably, K. marxianus DMKU 3-1042 as well as adapted strains exhibited stronger resistance to high temperatures (up to 48 ËšC) on agar plates [16]. It is assumed that only the surface cells of colonies on agar plates are exposed to oxidative stress, which increases with rising temperature [47], while individual cells in liquid media are directly exposed to the stress.” in the second paragraph of the discussion section.
- Cuny, C.; Lesbats, M.; Dukan, S. Induction of a global stress response during the first step of Escherichia coli plate growth. Appl Environ Microbiol. 2007, 73, 885-999.
How much Kluyveromyces marxianus DMKU 3-1042 were utilized to process? Furthermore, what are the conditions of reactions? Temperature, pH, ionic strength, for example. This information should be included in the abstract.
According to this comment, we have added the sentence of “In each cultivation step, 1% previous culture was inoculated into a medium containing 1% yeast extract, 2% peptone and 2% glucose, and cultivation was performed under a shaking condition” in the abstract.
- INTRODUCTION:
- Evolutionary Adaptation by Repetitive Long-Term Cultivation with Gradual Increase of Temperature for Acquiring Multi-Stress Tolerance and High Ethanol Productivity in Kluyveromyces marxianus DMKU 3-1042: This process needs to be explained in the introduction of the manuscript.
According to this comment, we have added the sentences of “Repetitive long-term cultivation with gradual increase of temperature (RLCGT) was performed by repeatedly cultivated yeast at a low temperature and then transferred it to a higher temperature and cultivated repeatedly in the same temperature. Yeast cells will adapt to the high temperature during they exposed with a stepwise increased temperature.” in the third paragraph of the introduction section.
- The Evolutionary Adaptation by Repetitive Long-Term Cultivation with Gradual Increase of Temperature for Acquiring Multi-Stress Tolerance and High Ethanol Productivity in Kluyveromyces marxianus DMKU 3-1042: What optimization strategy was used? Why was it used? This information needs to be explained in the introduction of the manuscript.
According to this comment, we have added the sentence of “Therefore, each cultivation step may optimize resistance to not only temperature but also other various stresses.” in the last paragraph of the introduction section.
- The new process presented were compared with a commercial material?? This information must be clear in the introduction.
According to this comment, we have added the sentence of “Therefore, HTF that uses stress-tolerant microbes can reduce the total cost of ethanol production, bring cheaper ethanol compared to commercial materials” in the second paragraph of the introduction section.
- What the advantages in Kluyveromyces marxianus DMKU 3-1042?
According to this comment, we have added one new reference [10] (Lertwattanasakul et al., 2022) in the first paragraph and added “like K. marxianus” in the second paragraph of the introduction section.
- Lertwattanasakul, N.; Nurcholis, M.; Rodrussamee, N.; Kosaka, T.; Murata, M.; Yamada, M. Kluyveromyces marxianus as a platform in synthetic biology for the production of useful materials. In Synthetic Biology of Yeasts; Darvishi Harzevili, F. (eds); Springer International Publishing, Cham, 2022; pp. 293-335.
These strategies used should be better explained in the manuscript.
According to this comment, we have added “, and provide benefits for industrial application [10, 21].” in the third paragraph of the introduction section.
- Nurcholis, M.; Lertwattanasakul, N.; Rodrussamee, N.; Kosaka, T.; Murata, M.; Yamada, M. Integration of comprehensive data and biotechnological tools for industrial applications of Kluyveromyces marxianus. Appl. Microbiol. Biotechnol. 2020,104, 475-488.
MATERIALS:
- Include the molar concentration of solutions.
According to this comment, we have included the molar concentration of acetic acid, formic acid, ethanol, vanillin, glucose, and xylose in the Materials and Methods section.
0.15 and 0.3% (v/v) acetic acid = 26.23 and 52.45 mM
0.1% (v/v) formic acid = 26.50 mM
8% (v/v) ethanol = 1.37 M
0.075, 0.1 and 0.15% (w/v) vanillin = 0.52, 0.70 and 1.05 mM
8, 12, 16 and 35% (w/v) glucose = 0.07, 0.10, 0.14 and 0.30 M
10% (w/v) xylose = 0.10 M
METHODS:
- Include the molar concentration of all the chemicals used, the way the methods are presented, not possible reproducibility.
- According to this comment, we have included the molar concentration of acetic acid, formic acid, ethanol, vanillin, glucose, and xylose in part of characterization of adapted strains in the Methods section, and we have also added the sentences of “This method introduces random mutations and cannot reproduce the same characteristics as the improved strains.” in evolutionary adaptation by RLCGT of the method section.
- Preparation procedure: Please include more details, temperature, pH, molar ratio, ionic strength.
According to this comment, we have included the molar concentration of acetic acid, formic acid, ethanol, vanillin, glucose, and xylose in part of characterization of adapted strains as described above. We also modified the statement of the Kluyveromyces marxianus DMKU 3-1042 pre-culture as follows.
“The pre-culture was prepared by transferring a single colony of 18 h culture on a YPD agar plate into 30 mL of YPD medium in a 100 mL Erlenmeyer flask, followed by incubation in a rotary shaker at 160 rpm for 18 h at 30 °C and. The pre-culture was inoculated into YPD medium or YP medium containing 160 g l−1 glucose at the initial optical density at 660 nm (OD660) of 0.1 and culture was carried out at 30 ËšC - 47 ËšC under a shaking condition at 100 or160 rpm."
- RESULTS AND DISCUSSION:
- The thermal stability to Kluyveromyces marxianus DMKU 3-1042 prepared is one of the most important application criteria for different applications. This stability depends on the preparation strategy. It also depends on the stabilization of the Kluyveromyces marxianus DMKU 3-1042. This discussion could be improved. Please include in the manuscript.
According to this comment, we have added the sentence of “Meanwhile, the thermal stability of K. marxianus DMKU 3-1042 is one of the most important application criteria for different applications, and the property is crucial for the preparation strategy of the strain or its stabilization.” in the second paragraph of the discussion section.
- The stability of Kluyveromyces marxianus DMKU 3-1042 its wide application in synthesis processes which nowadays are in a great demand from the point of view of industrial. Please include in the manuscript.
According to this comment, we have added the sentence of “The stability of K. marxianus DMKU 3-1042 is beneficial for a wide range of application in synthesis processes, which nowadays are in great demand from the point of view of industrial [10,21].” in the second paragraph of the discussion section.
- Did the authors conduct studies with non Kluyveromyces marxianus DMKU 3-1042? How can the authors state the effects of Kluyveromyces marxianus DMKU 3-1042 and non- Kluyveromyces marxianus DMKU 3-1042 on the characteristics of the produced biocatalysts? The authors need to explain the results in the manuscript.
According to this comment, we have added the sentences of “Recently, Candida tropicalis X-17 was subjected to RLCGT in the presence of a high concentration of glucose, and the resultant adapted strain increased ethanol production and improved multistress tolerance [48]).” in the third paragraph of the discussion section.
- Phommachan, K.; Keo-oudone, C.; Nurcholis, M.; Vongvilaisak, N.; Chanhming, M.; Savanhnaly, V.; Bounphanmy, S.; Matsutani, M.; Kosaka, T.; Limtong, S.; Yamada, M. Adaptive laboratory evolution for multistress tolerance, including fermentability at high glucose concentrations in thermotolerant Candida tropicalis. Energies. 2022, 15, 561.
- Was determined the full loading of Kluyveromyces marxianus DMKU 3-1042under the optimized conditions? This information must be clear in the manuscript.
We did not test other loadings of Kluyveromyces marxianus DMKU 3-1042 under the optimized conditions. We have thus changed the sentence from “1% of the culture was transferred into a fresh medium and cultivation was repeated under the same condition” to “1% of the culture was transferred into a fresh medium (Other inoculation sizes were not tested) and cultivation was repeated under the same condition” in evolutionary adaptation by RLCGT of the method section.
- The Kluyveromyces marxianus DMKU 3-1042 may experience aggregation (mainly near to the isoelectric point). This may be caused by undesired - interactions where inactivation that can stabilize incorrect in structures. This results must be cleared in the manuscript.
According to this comment, we have added the statement of “under a shaking condition, which provided oxygen and avoided aggregation of cells” in the last paragraph of the discussion section.
- The optimization of Kluyveromyces marxianus DMKU 3-1042 preparation process, the preparations shown having diffusion limitations? Considering the strategy presented in this manuscript. Please, this should be explained in the manuscript. What were the optimum conditions?
We did not try to optimization of K. marxianus DMKU 3-1042 preparation process but in the condition used in this study, a high level of the cell viability was kept from 12 h to 96 h at 30 ËšC (Figure 1), and thus we prepared pre-culture at 18 h at 30 ËšC.
- Others factors that cause the loss of durability and stability of the Kluyveromyces marxianus DMKU 3-1042 should be explained in the manuscript.
We described other factors that cause the loss of durability and stability of K. marxianus DMKU 3-1042 in the sentences of “Gradual rising temperature may reduce survival because in addition to temperature stress, higher temperature tends to accumulate more reactive oxygen species (ROS) that damage cells [5,21]. The combinations of various stresses including acetic acid, high temperatures, ROS, ethanol and other by-products are thought to influence the viability, resulting in enrichment of adapted mutants.” in the third paragraph of the discussion section.
- CONCLUSIONS
- The main contributions to the accomplishment of this work must be included in the conclusion. Please, authors must use numbers.
According to this comment, we have revised the sentences from “This study showed that RLCGT is an effective evolutionary adaptation procedure and provided evidence that adapted strains acquired have the ability to achieve high ethanol concentrations at high temperatures and are capable of tolerating multiple stresses, especially ACT strains, which are strongly resistant to acetic acid and formic acid. These beneficial properties are useful for industrial ethanol fermentation using lignocellulosic biomass as a substrate.” to “This study showed that (1) RLCGT is an effective evolutionary adaptation procedure and provided evidence that (2) adapted strains acquired have the ability to achieve high ethanol concentrations at high temperatures and (3) are capable of tolerating multiple stresses, especially ACT strains, which are strongly resistant to acetic acid and formic acid. These beneficial properties are useful for industrial ethanol fermentation using lignocellulosic biomass as a substrate.”
- Please, check all references according to the author's instructions.
We have checked all references.
Number of references
- No 9-19 were changed to be No 10-20
- No 20-44 were changed to be No 22-46
- No 45-50 were changed to be No 49-54
- Include more details in the figures (error bars) and tables captions.
We have improved to show clear error bars in figures 1 and 2 and added words in the caption of Table 1.
- The manuscript must be formatted according to the journal's standards.
We have improved the manuscript according to the format of this journal’s standards.
